# Assessing Disease Risks in Wildlife Translocation Projects: A Comprehensive Review of Disease Incidents

**DOI:** 10.3390/ani13213379

**Published:** 2023-10-31

**Authors:** Regina Kate Warne, Anne-Lise Chaber

**Affiliations:** School of Animal and Veterinary Sciences, The University of Adelaide, Mudla Wirra Road, Roseworthy, SA 5371, Australia

**Keywords:** conservation, disease risk analysis, reintroduction, relocation

## Abstract

**Simple Summary:**

There are many factors that can contribute to disease incursions in wildlife translocation projects. Through a systematic review of conservation translocation projects in literature, we found that the source of animals, the pathogen type, the host and the lack of disease risk analysis all contributed to disease as a result of the translocation. We recommend that future conservation translocations conduct comprehensive disease risk analyses and that a mandated database be established for the protocols and outcomes of all translocations to be published.

**Abstract:**

Although translocation projects have been instrumental in the supplementation or restoration of some wild populations, they also carry a large risk of disease transmission to native and translocated animals. This study systematically reviewed conservation translocation projects to identify projects that met the criteria for a translocation significant disease incursion (TSDI), whereby the translocation resulted in negative population growth rates or the failure of populations to grow due to an infectious disease—either in the native or translocated species. In doing so, risk factors for these incidents could be identified. Analysis of the resulting 30 TSDIs demonstrated that there was equal representation of TSDIs using wild-caught and captive-bred animals. Additionally, the type of pathogen predisposed in a TSDI was more likely a result of the animal group translocated (e.g., fungal pathogens were more likely to be detected in amphibian translocations) and it was nearly five times more likely for a disease to be encountered by a translocated species than for a disease to be introduced to a native population. However, there are numerous project-specific predisposing factors for TSDIs, and therefore it is essential that future translocation projects conduct thorough disease risk analysis as well as report their outcomes for the benefit of their own and future translocations.

## 1. Introduction

In the face of the Anthropocene, society is confronted with the task of conserving wildlife. To limit and reverse defaunation, it is important to address the drivers of species decline, including the following: overexploitation, habitat loss and modification, invasive species and diseases, pollution, and climate change [1]. However, the solutions to each of these issues are multifactorial, complex, and unlikely to be immediately achievable considering the number of stakeholders involved [1]. It therefore becomes important to conserve wild populations in the face of these threats.

For decades, animal translocation has been utilized as a conservation strategy, in which wildlife can be reintroduced to areas where they have been extirpated, be used to supplement existing wild populations, or be introduced to new areas. While translocations have been significant in the reintroduction of many species, including the Eurasian beaver (*Castor fiber)* to Scotland [2] and Przewalski’s horse (*Equus ferus przewalskii*) to China [3], not all translocations have been successful in establishing self-sustaining populations [4,5]. This has been due to predation, a lack of appropriate food, inappropriate environmental conditions, or, as will be the focus of this review, disease [4]. Disease poses a risk to translocated animals, native animals, and human handlers, and has been the cause of many conservation projects’ failure [5,6,7,8]. It is therefore important to anticipate and mitigate these disease risks via a ‘disease risk analysis’ [9]. The purpose of this study is to review the existing literature and determine which risk factors, if any, predispose conservation translocation projects to disease, including the role of disease risk analysis in these projects.

## 2. Materials and Methods

A comprehensive literature review was conducted by entering a Boolean search into the search engines Web of Science [10], Scopus [11], and CAB Abstracts [12] on 25 February 2023 using the following terms: ‘wildlife’, ‘endangered’, ‘animal’, ‘failure’, ‘unsuccessful’, ‘success’, ‘reintroduction’, ‘relocation’, ‘rewilding’, ‘disease’, ‘pathogen’, and ‘parasite’. The terms were searched for in the titles, abstracts, and keywords. The remaining articles were reviewed for reports or citations of wildlife translocations. If a particular paper cited a specific example of a translocation project and a disease incursion but did not provide sufficient details for the purposes of this study, further research was undertaken to supplement the information from the initial article.

### Project Eligibility

Translocation projects included in this study needed to meet the following criteria:Have ‘essential information’ readily available—whether through the primary article in the review or supplementary articles that were cited by the primary one. ‘Essential information’ included the following:a.The source of the animals to be translocated.b.The destination of the translocated animals.c.The year of the translocation.d.The translocated species.e.The species affected by disease because of the translocation.f.The suspected disease or pathogen.Be in any language.Be for the purpose of conservation, where the released animals were intended to either establish or supplement wild populations.Be any publication type (peer-reviewed or otherwise).Involve an infectious disease, whether officially diagnosed or merely suspected.Meet the criteria for a ‘translocation significant disease incursion’ (TSDI), whether this occurred following release or in captivity pre-release. A TSDI is hereby defined as a disease that occurs during or following the translocation of wildlife, whereby the consequences of which have a long-lasting negative effect on either (i) the translocated species or (ii) an endemic species at the translocation site. This can include the failure of the species to become established at the site or negative growth rates of the affected species.

Translocation projects were excluded from this study if any of the following occurred:The species failed to thrive due to reasons other than disease, i.e., predation, infertility, starvation.The project only translocated plant species.The project occurred prior to the initiation of the IUCN (1948).The project was an experiment studying the suitability of areas for translocation.

Eligible projects were then examined based on the species involved, the agent of disease (if known), the year of translocation, the location of translocation, the source of translocated animals, the direction of disease transmission, the suggested risk factors, and the completed risk analysis. For primary and supplementary papers, the veterinary qualifications of the authors were also specifically assessed through the listed credentials and affiliations on the publication.

All graphs were created using the RStudio ggplot package [13].

## 3. Results

The initial searches yielded a total of 400 articles, whereby 112 of which were duplicates. Following the full data review of the remaining articles, 26 articles cited a combined total of 30 projects that met the required criteria and are available in Appendix A. These projects are hereby classified as ‘TSDI’. To obtain the information required for the review, a further 18 supplementary articles were accessed.

### 3.1. TSDI Demographics

The studied projects were a mix of international and domestic projects when the number of countries involved in the translocation was identified (i.e., source and destination of the animals). Overall, 26.67% (*n* = 8) of TSDIs were international translocation projects, 66.67% (*n* = 20) were domestic, 3.33% (*n* = 1) were a mix of domestic and international sources, and 3.33% (*n* = 1) did not specify this information. Of the projects, 50% (*n* = 15) released captive-bred animals and 50% (*n* = 15) released wild-caught animals.

The temporality of TSDIs was varied when considering the year that the project commenced; 3.33% (*n* = 1) took place in the 1950s, 10.00% (*n* = 3) occurred in the 1970s, 23.33% (*n* = 7) were conducted in the 1980s, 23.33% (*n* = 7) took place in the 1990s, 26.67% (*n* = 8) occurred in the 2000s, and 13.33% were conducted in the 2010s.

The veterinary qualifications of the authors were mixed, and often indeterminable. Of the 44 papers, 9 were authored by at least one veterinarian while 11 had no veterinarian in the listed authors. A further six papers were authored by individuals working in veterinary faculties, with it being unclear whether they were registered veterinarians or not, and the veterinary qualifications of the authors of the remaining 18 papers could not be determined.

### 3.2. TSDI Event

Mammals were the most frequent animal group reported to be affected in TSDIs (50.00%; *n* = 15), followed by amphibians (23.33%; *n* = 7), birds (16.67%; *n* = 5), reptiles (6.67%; *n* = 2), and, finally, fish (3.33%; *n* = 1), as is indicated in Figure 1. The pathogen class implicated in the TSDI varied (Figure 1); 29.03% (*n* = 9) cited a parasitic cause, 25.81% were believed to be due to bacteria, 25.81% (*n* = 8) were believed to be due to a fungus, 16.13% (*n* = 5) were believed to be due to a virus, and 3.23% (*n* = 1) had an unknown cause. Four projects cited multiple pathogens that were believed to have contributed to the TSDI; three of these only involved one pathogen type (i.e., solely bacterial or solely parasitic), while one project cited both parasitic and viral involvement.

For the reviewed projects, most TSDIs involved the translocated animals encountering the disease upon release (76.67%; *n* = 23), but 16.67% of TSDIs (*n* = 5) were the result of animals contracting the infection in captivity prior to their release, 3.33% (*n* = 1) of TSDIs were due to the translocated animals introducing disease to native populations at the translocation site, and 3.33% (*n* = 1) had an unknown direction of transmission.

## 4. Discussion

### 4.1. TSDI Risk Factors and Their Mitigation

There are many factors that contribute to disease incursions, which have been simplified into the ‘epidemiological triad’ [14]. In this model, disease transmission relies on three pillars: the host, the agent (and its vectors), and the environment [14]. The risk factors of TSDIs have been separated into these three pillars.

#### 4.1.1. Host

For a host to become diseased, it must be susceptible to the agent of the disease. The host is more likely to become diseased when immunocompromised (i.e., stressed), naïve (i.e., individuals that are young or unexposed to the agent), or malnourished [15]. Interestingly, the proportion of TSDIs that came from captive-bred animals and the proportion that came from wild-caught animals were equal in this review. This therefore suggests that animals sourced from the wild remain at a similar risk of disease incursions upon release as animals that are bred in captivity. This is likely because, whether born in captivity or in the wild, if the animal is placed in a new environment, it will likely be naïve to the pathogens of that area [7]. Mammals were responsible for 50% of the TSDIs studied; however, it seems unlikely that mammals are more susceptible to disease than other animal groups. Instead, whether mammals are more often translocated and whether their relocations are more likely to be reported and consequently overrepresented in TSDI studies should be considered [4].

Vaccination can be a valuable strategy to build a host’s immunity to novel pathogens, although its application in translocations is controversial [16]. Furthermore, while vaccinations can reduce an individual’s disease susceptibility or severity, translocation projects are still vulnerable to disease incursions. One such example is Yellowstone wolves (*Lupus canis*). The project incorporated a comprehensive preventative health program in which all translocated wolves were vaccinated against canine parvovirus, canine distemper virus, canine adenovirus 1, leptospirosis, and canine parainfluenza virus, as well as treated with ivermectin, an anthelmintic [17,18]. Despite this, in the canine distemper outbreaks of 1995 and 2005, the Yellowstone populations suffered negative growth rates as the wild-born offspring of the translocated wolves were unvaccinated and therefore susceptible to this disease. However, the prevention of any mortality events and TSDIs in translocations, while ideal, is not always achievable [19]. Moreover, such incidents may not be detrimental to the overall success of the translocation. Therefore, preventative health regimes can be useful in allowing enough individuals to survive the outbreak. This was the case in the Yellowstone wolves which, despite the negative population growth in 1995, are currently an established population.

While the host’s exposure to pathogens is a vital predictor of susceptibility to disease [7], the influence of stress during translocation upon the incidence of disease cannot be underestimated [15]. Stress-induced immunosuppression was hypothesized to contribute to the development of yersiniosis and leptospirosis in beavers (*Castor fiber*) released in the Netherlands, as well as the development of yersiniosis in water voles released in the United Kingdom [20,21].

Ultimately, to maximize the health of all animals involved, it is critical to ascertain the diseases that may be harboured by the host and established populations and take appropriate measures to prevent disease, i.e., deworming, vaccination, selecting immunotolerant animals, low-stress handling, and controlled pathogen exposure [5,6,22]. Notably, this regime will be different for each translocation, where the inclusion of each component is decided through a risk–benefit analysis. For example, not all conservationists advocate for deworming protocols due to the risk of a fatal re-infestation and the potential for significant effects on the gut microbiome [23]. Crucially, researchers must consider the long-term effect of any interventions and if they will be sufficient for the overall establishment of the population, as in the Yellowstone wolves. In doing so, researchers can release animals that are less stressed and more immunocompetent and and healthy, thereby building their long-term resistance to pathogens that they may encounter [5,6]. Such actions should also protect native species due to the reduced pathogen shedding of released animals [7].

#### 4.1.2. Pathogens and Vectors

Naturally, the presence of pathogens and their vectors is a significant contributor to the transmission of disease. This review indicated that there was a similar proportion of TSDIs involving bacteria, parasites, and fungi, but it was less for viruses. However, the type of pathogen likely to lead to a TSDI seemed to be related to the animal group affected. For example, an amphibian TSDI was more likely to be caused by fungi, whereas a mammalian TSDI was more likely to be caused by parasites or bacteria (Figure 1). It therefore becomes important to anticipate and prevent common, species-specific agents rather than attempt to eliminate all diseases [9]. In doing so, the preventative health measures can be refined to target agents that are of the greatest concern, allowing for a more prudent use of resources and reducing the potential for disease.

However, even when researchers successfully predict the pathogen type likely to hinder a translocation, their strain and pathogenicity must also be ascertained. For example, 11 of the 26 Bighorn sheep (*Ovis canadensis*) translocated to the United States of America in 2015 died of pneumonia. These sheep had been vaccinated against *Mycoplasma* spp. but encountered a different strain (*Mycoplasma ovipneumoniae*) upon release, and therefore were susceptible to disease [24,25].

In the pursuit of comprehensive knowledge of the threats to animals in a translocation, researchers should utilize diagnostics both before, during, and after the release of wildlife, including the following: clinical examinations by qualified veterinarians, radiology, faecal examinations, haematology, serology, serum biochemistry, and necropsies, where appropriate [5,6,9]. Projects should also aim to report or publish their findings so that future researchers can anticipate the pathogens of concern. Furthermore, there should be comprehensive ecological studies on the proposed translocation sites so that potential vectors of disease can be identified.

#### 4.1.3. Environment

Environmental conditions also play a large role in the ability of a pathogen to cause disease. This is particularly clear given that more than 76% of TSDIs occurred due to translocated animals encountering disease upon release. As such, for researchers to appreciate the risk to translocated animals, disease surveillance of native populations should be conducted and reported on. Importantly, for TSDIs where the translocated species encountered the disease, not all articles commented on the native species. Therefore, the introduction of disease to native populations cannot be ruled out, especially given the lack of evidence of projects monitoring native species following the translocation.

The presence of asymptomatic carriers or contaminated waterways can be detrimental to the success of a translocation. One such case is the translocation of 27 alala (*Corvus hawaiiensis*) to Hawaii. Following their release, five died of toxoplasmosis and an additional number of animals were ‘sick’, with no official diagnosis being made [26]. The infections were suspected to be the result of the alala ingesting food contaminated by feral cats infested with *Toxoplasma* spp.

However, factors like humidity, temperature, population density, and pathogen load should also be assessed, as they can be crucial to the immunocompetence of animals and the viability of environmental pathogens. For instance, temperature was noted to affect the prevalence of *Lernaea cyprinacea* infestations in translocated Razorback suckers (*Xyrauchen texanus*) and Colorado squawfish (*Ptychocheilus lucius*), which contributed to the failure of these populations to become established [27].

Similarly, despite numerous attempts to reintroduce certain frog species in Australia, few have been successful due to the fungi *Batrachochytrium dendrobatidis* and *Batrachochytrium salamandrivorans* being ubiquitous in the environment. These two pathogens accounted for all amphibian TSDIs in this study and indicated that, as best as researchers may try, pathogens cannot be eliminated from the translocation site. Instead, it should be determined as to whether a proposed site of translocation is suitable at all, or if there are potential strategies available to increase an individual’s resilience to disease. For example, recent experiments revealed that environmental salinity protected amphibians against chytridiomycosis [28]. Therefore, in evaluating the intended translocation site’s ecological compatibility for the species and for diseases of concern, there may be improved outcomes for these projects.

### 4.2. The Importance of Disease Risk Analysis

Clearly, disease can greatly influence the outcome of translocations and therefore can significantly impair conservation attempts [9]. As such, for a translocation to occur, it should become mandatory for researchers to complete a disease risk analysis (DRA) in accordance with the IUCN prior to the commencement of a project. The IUCN ‘Guidelines for Wildlife Disease Risk Analysis’ [29] provide a framework in which to identify risk factors for disease exposure and transmission within a wildlife translocation project, as well as their potential management (Figure 2). These guidelines have only been available since 2014 and therefore were only available to 1 of the 30 TSDIs. Despite this, veterinarians are trained in the prevention of disease transmission [9,30] and could have been beneficial to translocation projects before formal instructions became available, yet only 9 of the 44 papers reviewed could be confirmed to be authored by at least one veterinarian. Therefore, the absence of animal health specialists and lack of industry guidance to conduct even an informal DRA could both predispose a project to a TSDI. As such, future translocations must involve direction from the IUCN guidelines and advice from veterinarians, among other professionals.

In the reviewed literature, most of the projects did not offer details of any risk analysis completed or any attempts to mitigate disease risks. Whether this is because a DRA was completed but not reported or communicated to the readers or because no DRA steps were taken is unknown. Where details were available, the projects had varying degrees of a DRA. This included some studies having knowledge of specific pathogens that may be encountered [17,18,31,32,33,34], vaccination [17,24,25,32], anthelmintic treatment [17,35], quarantine [32,35], pathogen testing [36], and veterinary examination [17,35,37].

In this study, 66% of TSDIs were domestic projects, thereby suggesting that translocations within the same country are more vulnerable to disease than international projects, which only accounted for 26% of TSDIs. Political and financial factors are major barriers to translocations [38], which may be considered to be greater in international projects. As a result of the greater ‘stakes’ of such projects, international translocations may be more likely to incorporate a DRA and therefore have fewer TSDIs. However, it should be considered as to whether domestic projects have a degree of complacency in whether native animals or habitats are ‘safe’, and therefore inadequate pre-release assessments are conducted. Although neither theories can be validated, they both emphasize the need for thorough disease risk analyses. In doing so, the potential for project success will be strengthened, irrespective of the species or the sites involved.

For example, in the reintroduction of the Eurasian beaver (*Castor fiber)* to Scotland, researchers identified the potential diseases that the beavers could transmit to wildlife, livestock, and humans upon release as well as assessed the risk of the beavers becoming infected with native diseases [2]. In considering the risk of diseases like tularaemia, rabies, and *Echinococcus multilocularis*, the researchers quarantined the beavers prior to their release, and during which they were screened for the diseases and parasites of concern. Post-release monitoring and public health surveillance indicated that the risk of disease from the translocation was negligible [2].

Notably, the completion of a DRA does not guarantee a disease-free translocation. In 2011, five European bison (*Bison bonasus*) were translocated from Poland to the Czech Republic. These bison were quarantined for 18 months, tested for parasitic infestations, and treated with anthelmintics. Despite their best efforts, the researchers postulated that the bison introduced *Ashworthius siderni* to native wild ruminants. It was hypothesized that this parasite could not be detected in quarantine as the *A.sidemi* undertook hypobiosis and it therefore was not excreted in the bison’s faeces and could not be tested [39]. In contrast, even if some form of a DRA is completed, the results may not be heeded. Prior to the release of Bongos (*Tragelaphus eurycerus*) in 2004, a feasibility study was undertaken, and experts recommended against the project taking place due to the significant disease risks anticipated. However, this advice was ignored and 5 of the 18 translocated Bongos died [5,7].

Therefore, while a DRA by no means prevents a TSDI, one should still be completed prior to all translocations—preferably with the assistance of a qualified veterinarian. Additionally, there should be careful consideration for non-infectious hazards that may predispose disease incursions or directly impede the establishment of populations through infertility or mortality [9,29]. This can include toxic, genetic, nutritional, handling, infrastructural, and metabolic factors, which should all be accounted for when conducting a DRA [29]. For example, multiple TSDIs cited the role of stress-induced immunosuppression in their respective disease incursions [20,21,32], and if the non-infectious factors that contribute to stress had been better understood and mitigated, the TSDI could have been avoided.

### 4.3. Further Recommendations

Identifying TSDIs in this systematic review was challenging, largely due to the insufficient project details available, in both primary and secondary sources. This highlights the importance for a systematic restructuring of the reporting process for translocation projects. As such, we recommend that, for the benefit of future translocations and other researchers, a formal, international database should be developed so that the procedures and outcomes of a particular translocation are publicly available. Within both this database and subsequent journal publications of the translocation, the following should be provided: the year of translocation, the translocated species, the source of the translocated species, the number of individuals translocated, the translocation site, the reason for translocation, the completed disease risk analysis (including the mitigation measures taken), the monitoring completed after release (including frequency of observation, the diagnostic tests used, and interventions), and the outcome of the project (including the population and reproductive rates of the translocated species, as well as the morbidity and mortality of both the translocated species and any native species that are believed to have been impacted by the translocation).

Moreover, while there are some suggested criteria to indicate a project’s ‘success’ [19], these do not appropriately assess disease incursions which have the potential to produce significant morbidities. Furthermore, even with these available criteria, projects still cannot be appropriately characterized as a ‘success’ or a ‘failure’ when the provided project details are incomplete. To allow for the appropriate assessment of disease in a translocation project, criteria should be developed that can effectively determine what level of disease at a translocation site is considered to be ‘significant’.

### 4.4. Limitations

In conducting this review, it was difficult to authenticate projects cited in the assessed literature. These included projects with unsatisfactory information available (whether through the primary source or supplementary sources) and the use of anecdotal evidence. Consequently, 12 projects could not be included in the final review and thereby a true representation of the risk factors for a TSDI is limited.

Additionally, in discussion with professionals involved in conservation, many translocations occur without the project being reported or published in journals (Warne pers comm). For those that are reported, post-release monitoring is not always completed and therefore the number of TSDI is likely vastly underestimated [19]. Accordingly, the small number of TSDI does restrict the generalizability of the results and their interpretations.

The obtained dataset only included cases that explicitly stated ‘disease’ as the cause of TSDI. However, as disease can predispose other causes of projects failing to establish healthy populations (predation, malnutrition, reproductive failures) it may be under-reported as a contributing cause. TSDI is most likely multifactorial, and therefore identifying its causes depends on the extent of investigation conducted within projects. Whether ill-health from infectious disease predisposed secondary biological problems eventually leading to the failure could not be ascertained.

This study did not assess the spatial or temporal influences upon disease incursions, both of which could affect TSDI incidence. The validity of diagnoses within these projects is also questionable given projects may have relied upon diagnostic tests (ELISA, PCR) or might have utilized more subjective measures (i.e., clinical examination, post-mortems, suspicion).

## 5. Conclusions

This investigation has highlighted that it is crucial that future translocation projects collaborate with relevant fields of expertise—including but not limited to—ecology, epidemiology, zoology, virology, and bacteriology, to conduct a comprehensive risk analysis to maximize the potential for success. It is also vital that all future projects collate and publish all data in relation to the studies, irrespective of their outcome, to guide subsequent endeavors. To maximize data collection and ongoing monitoring, investment into the development of surveillance technology is paramount. In focusing on these improvements, we welcome all researchers and stakeholders into the conversation, to learn, to improve and to build a world with thriving wildlife at its surface and ‘One Health’ at its core.

## Figures and Tables

**Figure 1 animals-13-03379-f001:**
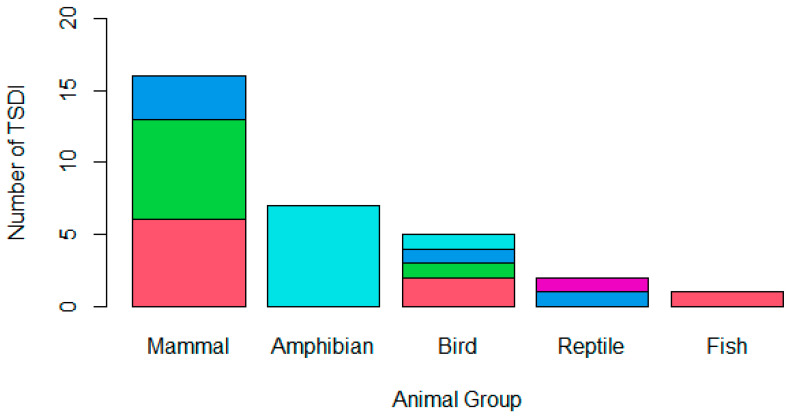
A stacked bar chart demonstrating the animal group affected by the TSDI amphibian (*n* = 7), bird (*n* = 5), fish (*n* = 1), mammal (*n* = 15), or reptile (*n* = 2) and the type of pathogen involved in the translocation, whether a parasite (pink), bacterium (green), virus (dark blue), fungus (light blue), or unknown (purple). One mammalian TSDI involved both parasitic and viral causes, while the remaining mammalian TSDI only involved one pathogen class; therefore, the chart demonstrates sixteen mammalian pathogen incursions across the fifteen TSDIs that occurred.

**Figure 2 animals-13-03379-f002:**
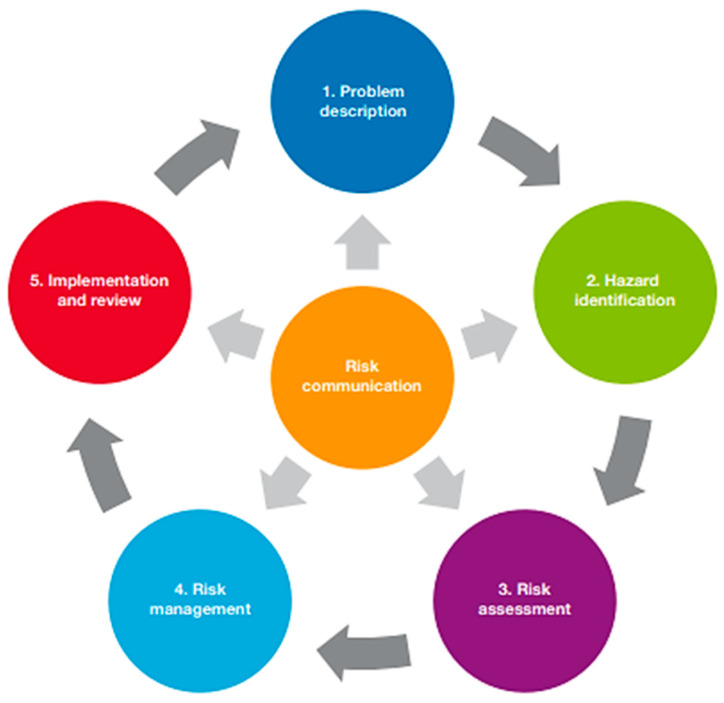
Steps in the disease risk analysis process [29].

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
