# Peer review of "Assessing Disease Risks in Wildlife Translocation Projects: A Comprehensive Review of Disease Incidents"

_animals, 2023, doi:10.3390/ani13213379_

Round 1

Reviewer 1 Report

Comments and Suggestions for Authors

Dear Authors,

Your manuscript represents a commendable and insightful exploration of a crucial facet within wildlife conservation, shedding light on the often-underestimated risks associated with translocation projects. Your research exhibits a well-structured and meticulously executed approach, making a significant contribution to our comprehension of disease transmission in translocated animal populations.

A particular strength of your manuscript lies in its comprehensive analysis, encompassing both wild-caught and captive-bred animals. This inclusion demonstrates that Translocation Significant Disease Incursions (TSDIs) affect both categories, similarly, emphasizing the necessity for a comprehensive assessment of disease risks, regardless of the animal source in translocation projects. Your identification of pathogen types associated with specific animal groups and the disparities in disease encounter rates for translocated species versus native populations are invaluable findings.

Your work offers an essential resource for individuals engaged in or interested in wildlife conservation and translocation projects. It underscores the urgency of adopting an enlightened and cautious approach to such initiatives, a critical factor for the enduring success of conservation efforts. I wholeheartedly concur with your recommendation for a standardized reporting format for all reintroduction or translocation project documents, promoting transparency and accessibility to the public, which can prevent the recurrence of errors.

Moreover, your manuscript is exceptionally well-written and easily comprehensible. I did not detect any grammatical errors. However, I have a few minor comments concerning Figures 1 and 2.

In Figure 1, it appears that the first bar exceeds the maximum value of 15, implying that the actual value of the bar may be 16 or 17.

With regards to Figure 2, although the content is clear, the image quality appears somewhat out of focus. Sharpening the image could enhance its clarity.

Lastly, in Appendix A, one minor clarification is necessary. The column labeled "Direction of transmission" lacks an explanation for the abbreviation "U." Providing this clarification would enhance the overall comprehensiveness of the document.

In conclusion, your work is truly commendable and makes a substantial contribution to the field.

Great job!

Author Response

We would like to thank the reviewer very much for their thoughtful response and support of our manuscript- it is much appreciated.

The range of the y-axis in Figure 1 has been edited as advised.

Likewise, Figure 2 has been sharpened.

The missing definition of 'U' in Appendix A has now been corrected.

Again, we would like to thank the reviewer for taking the time to provide us with this valuable feedback.

Reviewer 2 Report

Comments and Suggestions for Authors

In this well written manuscript there is discussion about infectious disease risks associated with wildlife translocation projects identified by review of the literature. Key risk factors are identified and it was noted that there were numerous project-specific predisposing factors for TSDI impacting local populations and translocated animals. While it is true that future translocation projects should require a specific disease risk analysis to prevent disease occurrence wildlife translocation efforts it would be good to discuss this further, are there good examples of where this has been beneficial ? For international animal translocations a risk assessment is often required as part of the importation process. Figure 2 is a bit hard to follow and could be improved, perhaps a flow chart with an example of a disease specific risk assessment ? The Table is appendix 2 is a nice summary but not all abbreviated letters used are outlined in the legend, it would be good to have this summary in the results section ?

Author Response

Thank you very much for the time taken to review our manuscript and for your thoughtful suggestions- we greatly appreciate it.

While it is true that future translocation projects should require a specific disease risk analysis to prevent disease occurrence wildlife translocation efforts it would be good to discuss this further, are there good examples of where this has been beneficial ?

Thank you for this suggestion, we have now revised our manuscript to include the success of the reintroduction of Eurasian beaver (Castor fiber) and how the disease-risk analysis facilitated the positive outcome.

Figure 2 is a bit hard to follow and could be improved, perhaps a flow chart with an example of a disease specific risk assessment ?

We appreciate the recommendation however we feel it is important to include the official figure used in the official IUCN 'Guidelines for Wildlife Disease Risk Analysis'.

The Table is appendix 2 is a nice summary but not all abbreviated letters used are outlined in the legend, it would be good to have this summary in the results section ?

Thank you for highlighting our oversight regarding the abbreviations- this has now been rectified. We aimed to summarise the contents of Appendix A in the 'Results' section as succinctly as possible and therefore left the greater detail to the Appendix for readers who sought more information.